

# Bayesian earthquake dating and seismic hazard assessment using chlorine-36 measurements (BED v1)

Joakim Beck[1], Sören Wolfers[1], and Gerald P. Roberts[2]

[1]Computer, Electrical and Mathematical Sciences & Engineering (CEMSE), King Abdullah University of Science and Technology (KAUST), Thuwal 23955-6900, KSA
[2]Department of Earth and Planetary Sciences, Birkbeck College, University of London, WC1E 7HX, UK

**Correspondence:** Joakim Beck (joakim.beck@kaust.edu.sa)

**Abstract.** Over the past twenty years, analyzing the abundance of the isotope chlorine-36 ($^{36}$Cl) has emerged as a popular tool for geologic dating. In particular, it has been observed that $^{36}$Cl measurements along a fault plane can be used to study the timings of past ground displacements during earthquakes, which in turn can be used to improve existing seismic hazard assessment. This approach requires accurate simulations of $^{36}$Cl accumulation for a set of fault-scarp rock samples, which are progressively exhumed during earthquakes, in order to infer displacement histories from $^{36}$Cl measurements. While the physical models underlying such simulations have continuously been improved, the inverse problem of recovering displacement histories from $^{36}$Cl measurements is still mostly solved on an ad-hoc basis. The current work resolves this situation by providing a MATLAB implementation of a fast, automatic, and flexible Bayesian Markov-chain Monte Carlo algorithm for the inverse problem, and provides a validation of the $^{36}$Cl approach to inference of earthquakes from the demise of the Last Glacial Maximum until present. To demonstrate its performance, we apply our algorithm to a synthetic case to verify identifiability, and to the Fiamignano and Frattura faults in the Italian Apennines in order to infer their earthquake displacement histories and to provide seismic hazard assessments. The results suggest high variability in slip rates for both faults, and large displacements on the Fiamignano fault at times when the Colosseum and other ancient buildings in Rome were damaged.

## 1 Introduction

A fundamental problem in earthquake science is the paucity of reliable earthquake records including multiple large magnitude earthquakes on individual faults. This hinders a more advanced understanding of earthquake recurrence, which is a prerequisite to forecasting future earthquakes. A promising approach to address this problem is in-situ chlorine-36 ($^{36}$Cl) cosmogenic exposure dating of active normal faults (Zreda and Noller, 1998; Mitchell et al., 2001; Schlagenhauf et al., 2010). This approach is based on the fact that earthquakes progressively exhume bedrock fault planes and thereby expose the bedrock to an increasing amount of cosmic radiation, which is the dominant source of $^{36}$Cl production in rock. The resulting characteristic $^{36}$Cl concentration profiles along fault planes therefore provide information about the timing and intensity of past earthquakes.

A comprehensive mathematical model of $^{36}$Cl production was provided in (Gosse and Phillips, 2001) and later formed the basis of a MATLAB code that computes $^{36}$Cl concentration profiles from temporal sequences of ground displacements (Schlagenhauf et al., 2010). Manual attempts to find best fits have subsequently been used for various faults (Benedetti et al.,



2002; Palumbo et al., 2004; Schlagenhauf et al., 2010, 2011; Yildirim et al., 2016). Manual fits, however, can be deceived by local minima and do not impart information on possible alternative solutions and the resulting uncertainties in the inference of earthquake histories. Indeed, the complexity of the model and the abundance of uncertain parameters results in a highly nonlinear and nonconvex problem, so that statistically reliable claims cannot be made without thorough modeling of prior

beliefs, and proper accounting for parameter and model uncertainties.

To address these issues, Cowie et al. (2017) used a Bayesian Markov-chain Monte Carlo (MCMC) sampler (Robert and Casella, 2004) to generate ensembles of plausible solutions. However, the candidate space considered was restricted by the assumption of equally spaced and sized displacements in active slip time periods. Also, priors on the placement and number of active time periods were not discussed. Furthermore, their approach did not incorporate uncertainties in important parameters

such as the $^{36}$Cl spallation and muonic production rates, the colluvial wedge mean density, which is difficult to measure in the field, and the timing of the demise of the last glacial maximum (LGM) after which slip was preserved on the fault plane, which is only known imprecisely. Finally, their implementation of the MCMC algorithm was not fully described, and no evidence of convergence was shown.

The new Bayesian MCMC method proposed in the current work adopts the Brownian motion model for earthquake recur-

rence (Matthews et al., 2002) to form a candidate space for earthquakes that arise and the associated prior probabilities. Besides improving the recurrence prior, this enables forecasts of future earthquakes, which is crucial for the subsequent task of regional seismic hazard assessment (Pace et al., 2006, 2014). Furthermore, we forgo the unrealistic assumption in (Cowie et al., 2017) that all displacements are of equal size and instead allow sizes to lie in a fault-dependent range. We employ parallel tempering (Woodard et al., 2009) to avoid premature convergence to local rather than global optima due to lack of explorative capabili-

ties. We verify global convergence with the diagnostic of Gelman and Rubin (1992). Since no model of $^{36}$Cl production can be completely accurate, we include an estimation of model discrepancy (Rougier, 2007; Rougier et al., 2013; Brynjarsdottir and O'Hagan, 2014), which results in more realistic confidence bands and may help guide various research efforts to improve the modeling of $^{36}$Cl accumulation. We also account for uncertainties in important parameters of the model that were previously held fixed. More specifically, we assume uncertain production rates, attenuation lengths, and colluvium density, and we infer

the time during the demise of the LGM at which the effect of erosion became outpaced by the ground displacements. We apply our algorithm to a synthetic case to verify the identifiability of past earthquakes in our model.

Finally, we study earthquake displacement histories for the Fiamignano and Frattura faults in the Italian Apennines. The results provide new evidence of slip-rate variability of normal faults in the Italian Apennines. The timing of slip-rate episodes can be reconciled with the historical record of earthquakes and damage to the Colosseum and other ancient buildings in Rome.

Beyond proposing a new approach to $^{36}$Cl earthquake dating, we extend our model to allow for Bayesian regional probabilistic seismic hazard assessment.

We supplement this paper with an easy-to-use MATLAB code of the proposed Bayesian MCMC method for earthquake dating and regional probabilistic seismic hazard assessment.



## 2  Simulation of $^{36}$Cl concentrations along fault scarps

Typical faults scarps are shown in Fig. 1a-c. These are characterized by two slopes that were originally joined, but are now offset across a geological fault due to surface slip events during earthquakes. The original planar slope (see Fig. 1d) was formed during the LGM, which for southern Europe ended around 20 000 to 12 000 years before present (20–12 ka), through intense

erosion on the upthrown side of the fault that was subject to freeze-thaw action (frost shattering) and sedimentation of the liberated slope debris (colluvium) on the downthrown side of the fault. Fig. 1d-f also shows how the morphology of the scarp changes through time across the LGM to post-glacial transition.

Before the demise of the LGM (see Fig. 1d) erosion via freeze-thaw action rapidly removed the surface uplifted by earthquakes (Allen et al., 1999; Peyron et al., 1998). During the demise the LGM, decreasing erosion rates may have allowed fault

slip to be preserved as a scarp, depending on the relative rates of erosion and fault slip. However, after the demise the LGM (Fig. 1e), the slip was preserved due to low erosion rates relative to the fault slip rates (Roberts and Michetti, 2004; Cowie et al., 2013, 2017). The type of slip is illustrated in Fig. 1h which shows decimeter scale slips produced during two earthquakes at the Mt. Vettore fault on 24 August 2016 ($M_w$ 6.1) and on 30 October 2016 ($M_w$ 6.6). The $M_w$ 6.6 surface rupture formed in 2–4 seconds during coseismic slip, recorded by Global Navigation Satellite System receivers placed either side of the fault

before the earthquake (Wilkinson et al., 2017).

The preserved fault plane can be sampled above ground and in excavated trenches. The $^{36}$Cl concentration in a fixed sample of limestone bedrock evolves according to

$$\frac{\mathrm{d}g(t)}{\mathrm{d}t} = \phi(t) - \lambda g(t), \tag{1}$$

where $g(t)$ is the $^{36}$Cl concentration ($\mathrm{atoms\,g^{-1}}$), $\phi(t)$ is the production rate ($\mathrm{atoms\,g^{-1}\,yr^{-1}}$) and $\lambda$ is the decay rate of $^{36}$Cl

($\mathrm{yr^{-1}}$). The *main* $^{36}$Cl production pathways in pure limestones are: spallation of $^{40}$Ca; muon capture by $^{40}$Ca, and thermal neutron capture by $^{35}$Cl. These processes depend on cosmic radiation flux which is attenuated by the surrounding environment composed of air, colluvium and rock (see Fig. 1g). As a consequence, the production rate $\phi(t)$ in a rock sample through time is strongly influenced by earthquake-induced changes to the surrounding environment. Roughly speaking, samples taken from the trench have experienced $^{36}$Cl production at low rates due to shielding by overlying colluvium and neighboring limestone

bedrock, see Fig. 1g, whereas above-ground samples have experienced early sub-surface production before exhumation as well as subsequent surface production. This results in a characteristic $^{36}$Cl concentration profile along the fault plane, which captures its history of ground displacements. Note that samples must be taken from a fault plane away from post-glacial alluvial fans and eroded gullies where exposure is also influenced by post-glacial erosion and sedimentation (see Fig. 1f)

Formulae for surface production through the aforementioned processes as well as the associated attenuation, or shielding,

factors can be found in (Gosse and Phillips, 2001; Schlagenhauf et al., 2010). More than a decade of contributions to the modeling of $^{36}$Cl production (Lal, 1988; Phillips et al., 1996; Mitchell et al., 2001) have been assembled into a MATLAB code that computes $^{36}$Cl concentrations in a set of bedrock samples for given sequences of scarp displacements and event times; the code is provided in the supplement of (Schlagenhauf et al., 2010). The code solves Eq. (1) by a first-order finite-difference



**Figure 1.** Images and illustrations of scarp evolution and radiation environment





scheme, and uses various simplifications for calculating the shielding factors; for example, it assumes that the cosmic ray flux decays exponentially with the depth of a sample beneath the colluvial wedge.

As part of this work, we provide a MATLAB code that circumvents some of the approximations and simplifications of (Schlagenhauf et al., 2010), in particular in the computation of shielding factors. We use exact solutions of Eq. (1), piece-
wise exponentials, which is possible under the assumption of a constant-in-time production rate. These changes improved the predicted $^{36}$Cl concentration by around 5% in our numerical experiments. Furthermore, we introduce an offline phase during which we pre-compute a database of shielding factors for a *sparse grid* (Barthelmann et al., 2000) of possible values of the model parameters. By interpolating between these factors, we were able to accelerate the computations during the inverse problem by two orders of magnitude. Finally, an improvement of particular relevance to our case studies is that we also consider
events before the end of the LGM. We approximate the effects of the LGM on erosion in a binary manner, by assuming a single point in time, $T_{\mathrm{init}}$, before which erosion immediately eroded scarps formed by earthquakes, and after which erosion stopped completely.

The provided MATLAB code calculates $^{36}$Cl concentrations in a set of bedrock samples for given sequences $\boldsymbol{d} = (d_1, \ldots, d_N)$ and $\boldsymbol{s} = (s_1, \ldots, s_N)$ of scarp displacements and event times and the following fault site properties:

– **Geometric description (see Fig. 1g):**

  – Dip of the lower slope, $\alpha$ (°)

  – Fault plane dip, $\beta$ (°)

  – Dip of the upper slope above the fault, $\gamma$ (°)

– **Geological properties:**

– Rock mean density, $\rho_{\mathrm{rock}}$ ($\mathrm{g\,cm^{-3}}$)

  – Colluvial wedge mean density, $\rho_{\mathrm{coll}}$ ($\mathrm{g\,cm^{-3}}$)

  – Spallation production rate of $^{36}$Cl by fast secondary neutrons at the surface from $^{40}$Ca, $\Psi_{\mathrm{sp}}$ ($\mathrm{atoms\,g^{-1}\,yr^{-1}}$)

  – Slow muon capture production rate of $^{36}$Cl at the surface from $^{40}$Ca, $\Psi_{\mu}$ ($\mathrm{atoms\,g^{-1}\,yr^{-1}}$)

  – Neutron apparent attenuation length, $\Lambda_{\mathrm{sp}}$, and muon apparent attenuation length, $\Lambda_{\mu}$, for a horizontal unshielded
surface (cm)

  – Chemical compositions (ppm) of the rock in each sample and of the soil and pebbles in the colluvium

– **Further properties:**

  – The influence of the geomagnetic field is specified through scaling factors for fast neutrons and slow muons, see the supplementary material of (Schlagenhauf et al., 2010)

– The time during the demise of the LGM at which the effect of erosion became outpaced by the ground displacements, $T_{\mathrm{init}}$ (yr)



## 3 Bayesian inference of displacement histories using MCMC

In this section, we present a Bayesian MCMC algorithm that solves the inverse problem of the previous section, i.e., the inference of past earthquakes from $^{36}$Cl measurements $\boldsymbol{y} = (y_1, \ldots, y_M)$ at $M$ sample sites along a single fault scarp. More specifically, we compute posterior distributions for the vectors of fault displacements and event times, $\boldsymbol{d} = (d_1, \ldots, d_N)$ and $\boldsymbol{s} = (s_1, \ldots, s_N)$, respectively (and in particular for the number of events, $N$). In addition, we treat the values of $\boldsymbol{z} := (T_{\text{init}}, \Psi_{\text{sp}}, \Psi_\mu, \Lambda_{\text{sp}}, \Lambda_\mu, \rho_{\text{coll}})$ as uncertain parameters and include them in the inference problem.

Denoting by $\boldsymbol{y}^*$ the vector of *true* $^{36}$Cl concentrations, and by $\boldsymbol{g}(\boldsymbol{x})$ the output of the computer model of Eq. 2 for given values of $\boldsymbol{x} := (\boldsymbol{d}, \boldsymbol{s}, \boldsymbol{z})$, we first observe the equation

$$\boldsymbol{y} = \boldsymbol{y}^* + \boldsymbol{\epsilon} = \boldsymbol{g}(\boldsymbol{x}) + \boldsymbol{\delta} + \boldsymbol{\epsilon} \tag{2}$$

where $\boldsymbol{\epsilon}$ is the measurement error, i.e., the discrepancy between the true and the measured concentrations, and $\boldsymbol{\delta} := \boldsymbol{y}^* - \boldsymbol{g}(\boldsymbol{x})$ is the model error, i.e., the discrepancy between the model output and the true concentrations (Rougier et al., 2013; Brynjarsdottir and O'Hagan, 2014). We assume that the measurement errors are independent and normally distributed, $\boldsymbol{\epsilon} \sim \mathcal{N}(0, \boldsymbol{\sigma}^2)$ for a given vector $\boldsymbol{\sigma} = (\sigma_1, \ldots, \sigma_M)$ of positive real numbers. The model error is assumed to be proportional to the measurements, $\boldsymbol{\delta} \sim \mathcal{N}(0, (\rho \boldsymbol{y})^2)$, and we include the value of $\rho > 0$ in the inference problem.

To describe the Bayesian inference method, let us denote by $\boldsymbol{\theta} := (\boldsymbol{x}, \rho, \boldsymbol{\phi})$ the vector of all unknowns, where $\boldsymbol{\phi}$ contains auxiliary variables that are described in Sect. 3.1 below. If we have a prior belief on the value of $\boldsymbol{\theta}$, described by a probability distribution $P_{\boldsymbol{\theta}}$, then the posterior distribution for $\boldsymbol{\theta}$ given the measurements $\boldsymbol{y}$ can be found using Bayes' rule,

$$P_{\boldsymbol{\theta}|\boldsymbol{y}} \propto P_{\boldsymbol{y}|\boldsymbol{\theta}} P_{\boldsymbol{\theta}}, \tag{3}$$

with the distribution of $\boldsymbol{y}$ conditioned on a fixed value of $\boldsymbol{\theta}$, or *likelihood*, given by

$$P_{\boldsymbol{y}|\boldsymbol{\theta}} \sim \mathcal{N}(\boldsymbol{g}(\boldsymbol{x}), \boldsymbol{\sigma}^2 + (\rho \boldsymbol{y})^2),$$

To gain information about $P_{\boldsymbol{\theta}|\boldsymbol{y}}$, we employ a MCMC method (Robert and Casella, 2004), which generates samples that can be used to approximate statistical properties of $P_{\boldsymbol{\theta}|\boldsymbol{y}}$. More specifically, we use a *Metropolis-Hastings* MCMC method, which generates a sequence (chain) $(\boldsymbol{\theta}_k)_{k=1}^\infty$ of random samples, where an initial sample is taken from the prior distribution and each successive sample is generated from its predecessor by means of *random proposal functions* and an acceptance step that guarantees that the sample distribution converges to the desired distribution $P_{\boldsymbol{\theta}|\boldsymbol{y}}$ as $k \to \infty$. To accelerate this convergence, we employ a parallel tempering approach that simultaneously generates $L > 1$ chains $(\boldsymbol{\theta}_k^{(l)})_{k=1}^\infty$, $1 \le l \le L$ with progressively flattened likelihood distributions,

$$P_{\boldsymbol{y}|\boldsymbol{\theta}}^{(l)} \sim \mathcal{N}(\boldsymbol{g}(\boldsymbol{\theta}), \kappa_l(\boldsymbol{\sigma}^2 + (\rho \boldsymbol{y})^2)), \quad 1 = \kappa_1 < \cdots < \kappa_l < \cdots < \kappa_L,$$

and randomly swaps states between neighboring chains such that the resulting samples of the first chain $(\boldsymbol{\theta}_k^{(1)})_{k=1}^\infty$, which uses $\kappa_1 = 1$, are still distributed according to $P_{\boldsymbol{\theta}|\boldsymbol{y}}$ (in the limit). This has been shown to accelerate the exploration of the state space in cases where the posterior distribution has multiple local maxima (Woodard et al., 2009).





To fully specify our inference method, it remains to describe the prior distribution $P_{\boldsymbol{\theta}}$ and the proposal functions that are used for sample generation.

### 3.1 The prior distribution $P_{\boldsymbol{\theta}}$

We describe the prior distributions of different components of $\boldsymbol{\theta}$ with the understanding that separately described components
are assumed to be stochastically independent.

For the parameter $\rho$, which controls the relative model error, we assume a uniform prior distribution, $\rho \sim \mathcal{U}[0, \rho_{\max} = 0.1]$.

To describe a prior for the earthquake times $\boldsymbol{s}$ (and in particular the number of events $N$), we extend our state space by a stochastic process in $[-T_{\min}, 0]$ that is used to model earthquake occurrence, see Fig. 2. In doing so, we follow (Matthews et al., 2002), where the time between successive earthquakes is modeled by an inverse Gaussian distribution, which is also called the
Brownian passage-time distribution since it describes the time required by a Brownian motion to reach a certain threshold. To account for large-scale time periods with differing average earthquake frequencies, we extend the model in (Matthews et al., 2002) by including a Poisson($\Lambda \, T_{\min}$)-distributed number $J$ of switch points $(t_j)_{j=1}^{J}$, where $t_j \sim \mathcal{U}([-T_{\min}, 0])$ and $\Lambda \sim \mathcal{U}([10^{-4}, 10^{-3}])$. This is equivalent to the assumption that the times between successive switch points are exponential random variables whose mean $1/\Lambda$ is inferred in the interval $[10^3, 10^4]$. We then define the drift $a(t)$ and volatility $b(t)$ of the
Brownian motion in the intervals of the resulting partition of $[-T_{\min}, 0]$ by

$$(a(t), b(t)^2) := (1/\nu_j, \tau^2/\nu_j) \text{ for } t_j \le t \le t_{j+1} \text{ (with } t_0 := -T_{\min} \text{ and } t_{J+1} := 0),$$

where the average inter-arrival times $\boldsymbol{\nu} = (\nu_j)_{j=0}^{J}$ are independent and identically distributed according to an Inverse Gamma distribution with mean $m \sim \mathcal{U}([200, 2000])$ and shape parameter $\alpha \sim \mathcal{U}([1, 10])$, and where $\tau \sim \mathcal{U}([0, 1])$ controls the short-scale recurrence variability (with the above definitions, the standard deviation of inter-arrival times in the $j$-th subinterval
is $\tau \nu_j$). Numerically, given values of $\boldsymbol{\nu}$ and $\tau$, we use forward Euler-Maruyama time stepping (Higham, 2001) with time discretization $\Delta t \approx 15\,\mathrm{yr}$ to simulate the resulting stochastic process on $[-T_{\min}, 0]$, with the modification that we reset the process to 0, and generate an earthquake by extending $\boldsymbol{d}$ and $\boldsymbol{s}$, each time it reaches the threshold 1. In formulae, we let

$$\hat{X}_{i+1} := X_i + a(i\Delta t)\Delta t + b(j\Delta t)W_{i+1},$$

$$X_{i+1} := \begin{cases} \hat{X}_{i+1} & \text{if } \hat{X}_{i+1} < 1 \\ 0 & \text{else} \end{cases} \tag{4}$$

The values $W_i \sim \mathcal{N}(0, \Delta t)$ together with $\boldsymbol{\nu}$ and $\tau$ and the initial value $X_0 \sim \mathcal{U}([0, 1])$ form the vector of auxiliary variables $\boldsymbol{\phi}$
referred to above, which fully determines the process $(X_i)_{i=0}^{T_{\min}/\Delta t}$, which in turn determines the vector of earthquake times $\boldsymbol{s}$.

For the prior distribution of the displacement vector $\boldsymbol{d}$ conditioned on the number $N$ of earthquake events, we use a uniform prior on the hypercube $[d_{\min}, d_{\max}]^N$ conditioned on the requirement that $\sum_{n=1}^{N} d_n = H_{\mathrm{sc}}$, where $H_{\mathrm{sc}}$ (see Fig.1) is the present height of the fault scarp, and $d_{\min}$ and $d_{\max}$ are fault-dependent bounds on displacement sizes.

Finally, we assign prior distributions to the components of $z$: $T_{\mathrm{init}} \sim \mathcal{U}([12\,000, 20\,000])$, $\Psi_{\mathrm{sp}} \sim \mathcal{N}(48.8, 1.7)$, $\Psi_{\mu} \sim \mathcal{N}(190, 19)$,
$\Lambda_{\mathrm{sp}} \sim \mathcal{U}([180, 220])$, and $\Lambda_{\mu} \sim \mathcal{U}([1300, 1700])$, where the first three are based on results of (Allen et al., 1999), (Stone et al.,





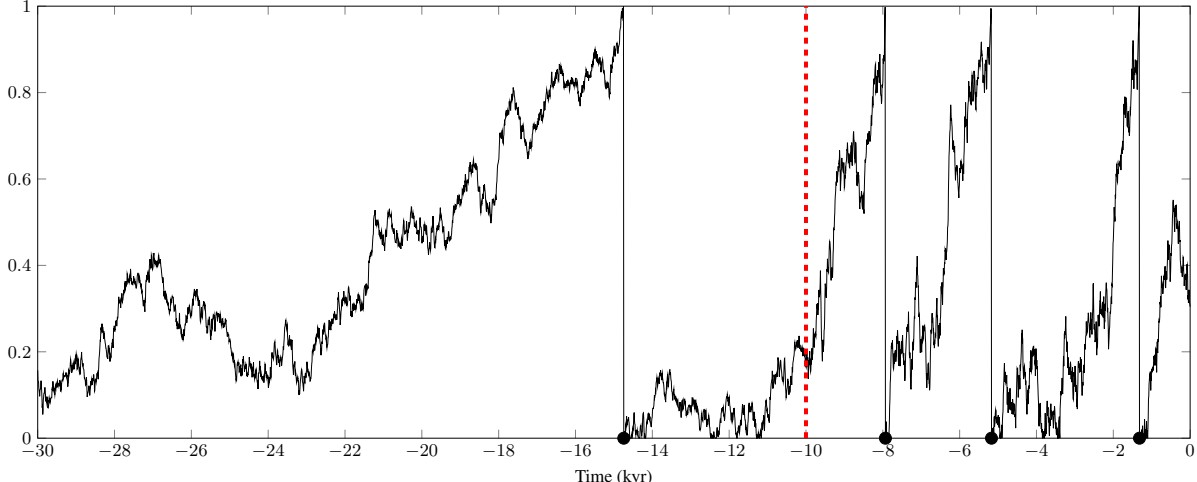

**Figure 2.** Sample path of stochastic process used for earthquake event time generation. Sample path generated with $X_0 = 0.16$, $T_{\min} = -30\,000\,\text{yr}$, $t_1 = -10\,000\,\text{yr}$, $\boldsymbol{\nu} = (15\,000\,\text{yr}, 3000\,\text{yr})$, $\tau = 0.5$, and $\Delta t \approx 7\,\text{yr}$. The corresponding earthquake times are represented by dots on the time axis, the switch point $t_1$ is represented by a red vertical line.

1996), and (Heisinger et al., 2002), respectively, and the remaining are chosen to be uniform around values taken from (Schlagenhauf et al., 2010). It is difficult to accurately measure the value of $\rho_{\text{coll}}$ in the field due to compaction of the sediment during excavation of the sample trench at the base of the fault scarp, so we adopt the relatively wide prior $\rho_{\text{coll}} \sim \mathcal{U}([1.2, 1.8])$ for all case studies considered in this work.

## 3.2 MCMC proposal functions

To explore the state space, we design a number of proposal functions and apply a subset of these before each rejection step in the MCMC algorithm. For each component of $\boldsymbol{z}$ as well as for the values of $\rho$, $\boldsymbol{\nu}$, $\tau$ and $X_0$, we include a global proposal from the corresponding prior distribution as well as a local proposal based on a normal distribution around the current value. To propose the partition of $[-T_{\min}, 0]$ and the corresponding piecewise constant drift and volatility coefficients, we employ Reversible Jump MCMC (Green, 1995), which allows for the application of MCMC to variables whose state space contains subspaces of different dimensionality. To propose new values of $(W_i)_{i=1}^{T_{\min}/\Delta t}$ that drive the process $(X_i)_{i=1}^{T_{\min}/\Delta t}$, we again use a global proposal, which redraws all values independently, as well as a Brownian bridge-type local proposal that redraws the values within a random subinterval of $[-T_{\min}, 0]$ from their prior distribution conditioned on maintaining their sum. If a proposal changes the number of earthquakes, the earthquake displacement vector $\boldsymbol{d}$ (which then has a different size) is re-sampled too, and if a proposal changes the number of switch points, the vectors of drift and volatility coefficients are re-sampled as well.



# 4 Application to synthetic $^{36}$Cl data

In this section we apply our Bayesian MCMC method to synthetic $^{36}$Cl data. To generate these data, we drew $d$ and $s$ from the prior distributions described in Sect. 3.1 with $d_{\min} = 10$ and $d_{\max} = 110$ and applied the computer model of Sect. 2 using the chemical compositions of 146 rock samples from the Fiamignano fault (Cowie et al., 2017) and the values

$z = (T_{\text{init}} = -19000, \Psi_{\text{sp}} = 49.5, \Psi_\mu = 200, \Lambda_{\text{sp}} = 195, \Lambda_\mu = 1700, \rho_{\text{coll}} = 1.6)$. The remaining parameters, which are not part of the inference problem, were chosen as $H_{\text{sc}} = 2705$, $H_{\text{tr}} = 115$, $\rho_{\text{rock}} = 2.7$, $\alpha = 23$, $\beta = 42$, and $\gamma = 33$, based on the true values of the Fiamignano fault that were measured in the field. Finally, we perturbed the $^{36}$Cl concentration values according to Eq. 2, with standard deviation $\sigma = 2.5 \times 10^{-2} g(x)$ for the measurement error and $\rho = 3 \times 10^{-2}$ in the model error term. The realizations of $d$ and $s$ are given in the supplementary material.

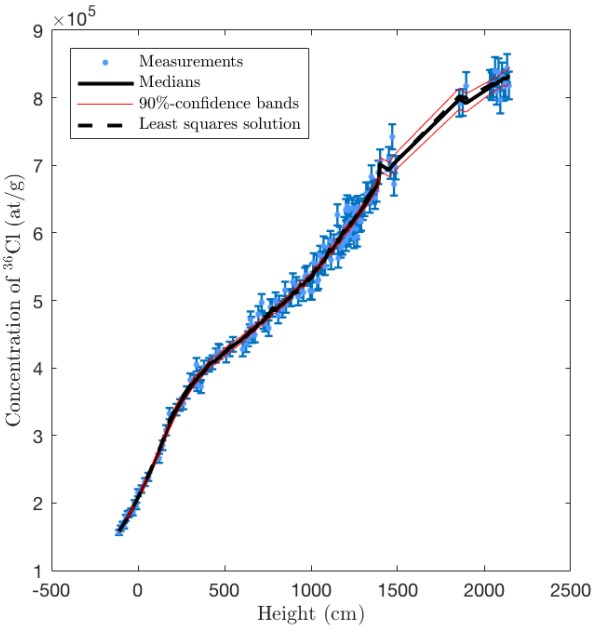

**Figure 3.** Measured and sampled $^{36}$Cl concentrations for the synthetic case with 146 $^{36}$Cl measurements

We ran our MCMC method run using the priors specified in Sect. 3.1 for the uncertain parameters, i.e., $T_{\text{init}} \sim \mathcal{U}([12\,000, 20\,000])$, $\Psi_{\text{sp}} \sim \mathcal{N}(48.8, 1.7)$, $\Psi_\mu \sim \mathcal{N}(190, 19)$, $\Lambda_{\text{sp}} \sim \mathcal{U}([180, 220])$, and $\Lambda_\mu \sim \mathcal{U}([1300, 1700])$, $\rho_{\text{coll}} \sim \mathcal{U}([1.2, 1.8])$.

The results presented in this section are based on 2 independent MCMC chains, each consisting of $L = 20$ parallel tempering levels. We performed $1\,374\,462$ MCMC iterations of each chain, resulting in a combined number of $2\,474\,032$ samples in the first levels of the two independent chains after a $10\%$ burn-in period. Among these samples were $843\,735$ distinct scenarios,

whose repetitions correspond to rejected proposals and reflect their statistical weight. Finally, to accelerate post-processing and to decrease memory consumption, we saved only each 5th scenario together with its number of repetitions (*thinning*).





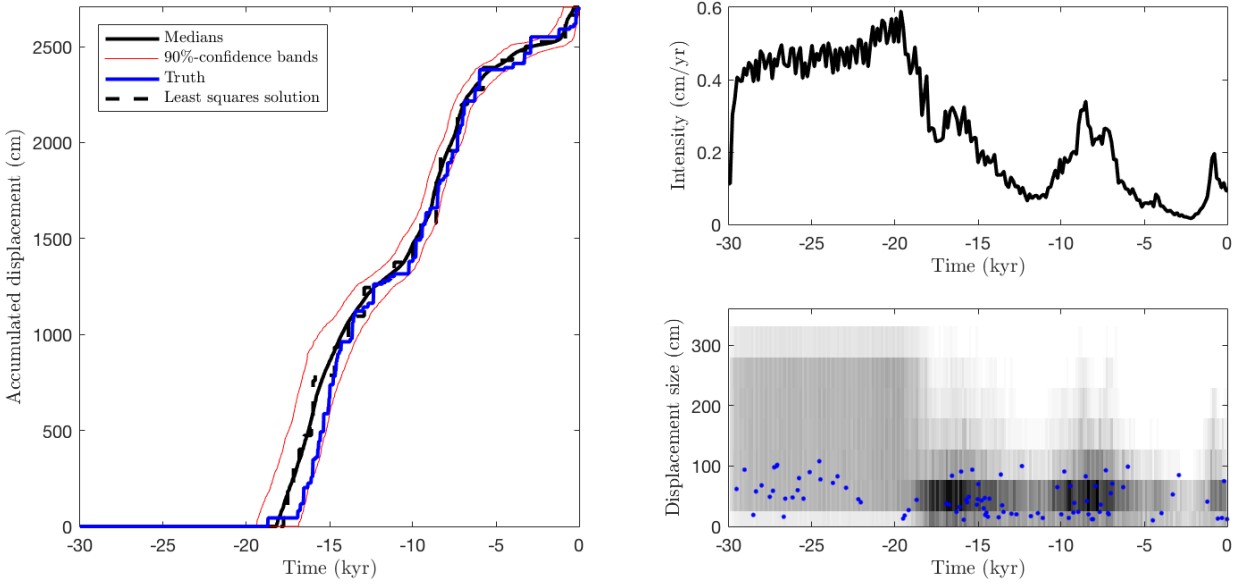

**Figure 4.** The synthetic case with 146 $^{36}$Cl measurements: the accumulated displacement (left), the mean earthquake intensity (right top) and an event scatter plot (right bottom) showing true events (blue) and events from posterior samples (with grey scale to indicate frequencies).

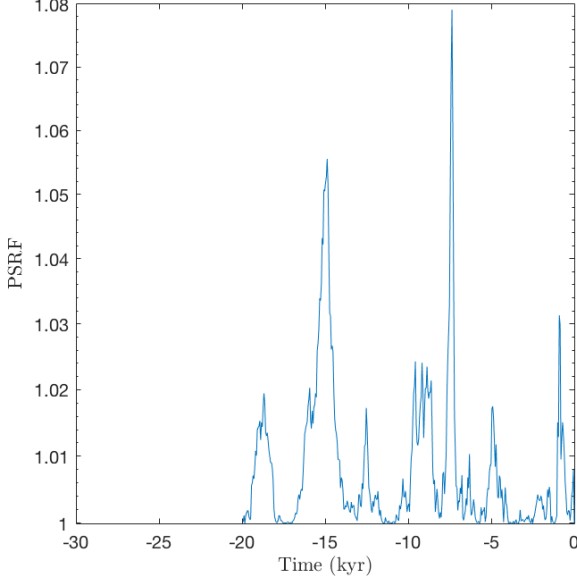

**Figure 5.** Gelman-Rubin diagnostic for the synthetic case with 146 $^{36}$Cl measurements





Figure 3 shows that the medians of the $^{36}$Cl concentrations of the posterior samples provide a good fit to the synthetic measured concentrations. More importantly, Figure 4 (left) shows that the posterior median of the accumulated displacement is close to the true value throughout the entire time span. This indicates that past earthquake activity can be recovered despite measurement errors (here 2.5%), model errors (here 3%) and parameter uncertainties. Figure 4 (right) shows the posterior

5    mean earthquake intensity, i.e., the average displacement per time (computed for bins of width ~500 years), and a scatter plot of all earthquake events of the posterior samples.

In both plots, the erosion during the last LGM has removed information and so leads to almost uniform posterior intensities across time and through event sizes during the LGM, which means that individual events in that period cannot be recovered. Nevertheless, including such events in our approach can be considered a way to obtain realistic prior $^{36}$Cl concentrations at the

10   end of the LGM, $T_{\text{init}}$, as opposed to the alternative to start with zero concentrations or imposing ad-hoc pre-exposure times (Schlagenhauf et al., 2010).

Figure 5 shows that the PSRF value associated with the accumulated displacement values of Figure 4 is below 1.08.

Since the collection and chemical analysis of $^{36}$Cl samples involve time-consuming and costly fieldwork and lab work, in particular the AMS analysis, it is worthwhile knowing whether fewer than 146 samples can provide similar insights. To answer this question, we repeated our computations with a subset of 16 rock samples (see Fig. 6). Figure 7 shows that this reduced

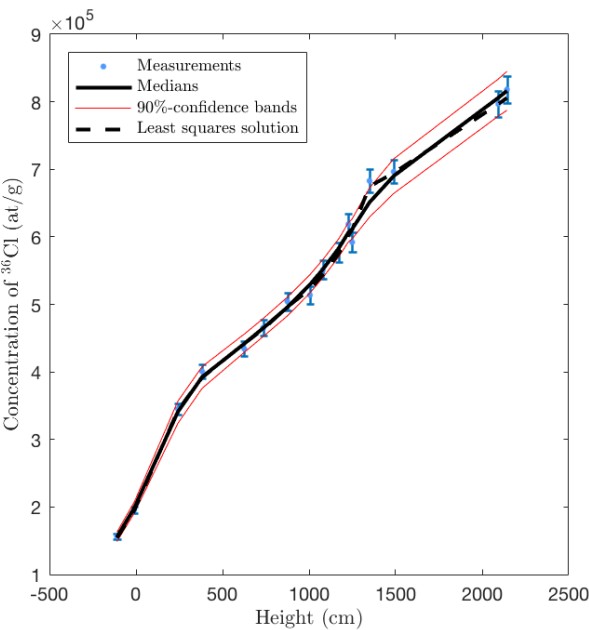

**Figure 6.** Measured and sampled $^{36}$Cl concentrations for the synthetic case with 16 $^{36}$Cl measurements

dataset leads to similar results as the complete dataset (cf. Fig. 4). These results are based on 421 022 MCMC iterations of each chain and using the same number of chains, burn-in and thinning as before, which led to a maximal PSRF value of 1.02.





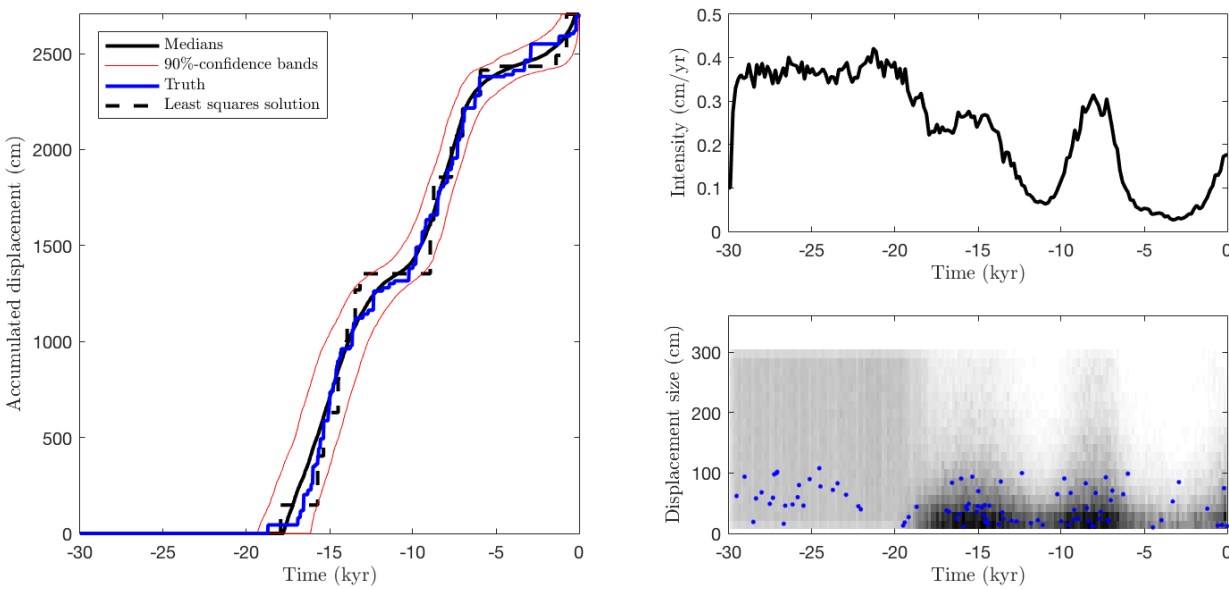

**Figure 7.** The synthetic case with 16 $^{36}$Cl measurements: the accumulated displacement (left), the mean earthquake intensity (right top) and an event scatter plot (right bottom) showing true events (blue) and events from posterior samples (with grey scale to indicate frequencies).

# 5 Application to normal faults in the Italian Apennines

The Italian Apennines contain many examples of bedrock scarps on active normal faults and it has been suggested that rates of slip produced by repeated earthquake rupture since the LGM can be used to investigate seismic hazard and the mechanics of continental deformation (Piccardi et al., 1999; Roberts and Michetti, 2004; Cowie et al., 2013). The active normal faults

work to extend the continental crust at the present day in formerly-thickened crust of the Alpine-Apennines collision zone. The extension started at 2–3 Ma, as evidenced by dated sediments in extensional sedimentary basins formed by fault activity (Cavinato and Celles, 1999; Roberts et al., 2002). The basins occur on the downthrown side of the faults, and contain accumulations of sedimentary layers that are usually hundreds to a few thousand meters thick. When added to the offset implied by the uplifted mountains on the upthrown side of the faults, total fault offsets are up to 2–2.5 km, which means long-term rates

of vertical motion across the faults are in the order of 0.1 cm/yr.

In this section we apply our method to the Fiamignano and Frattura faults of the Italian Apennines, and show its applicability to regional probabilistic seismic hazard assessments.

## 5.1 Fiamignano fault

The Fiamignano fault is located on the southwest (SW) flanks of the Apennines, approximately 60 km northeast of Rome.

Based on a study of 34 damaging earthquakes recorded in (Guidoboni et al., 2007; Galli and Molin, 2014; Vittori, 2015),




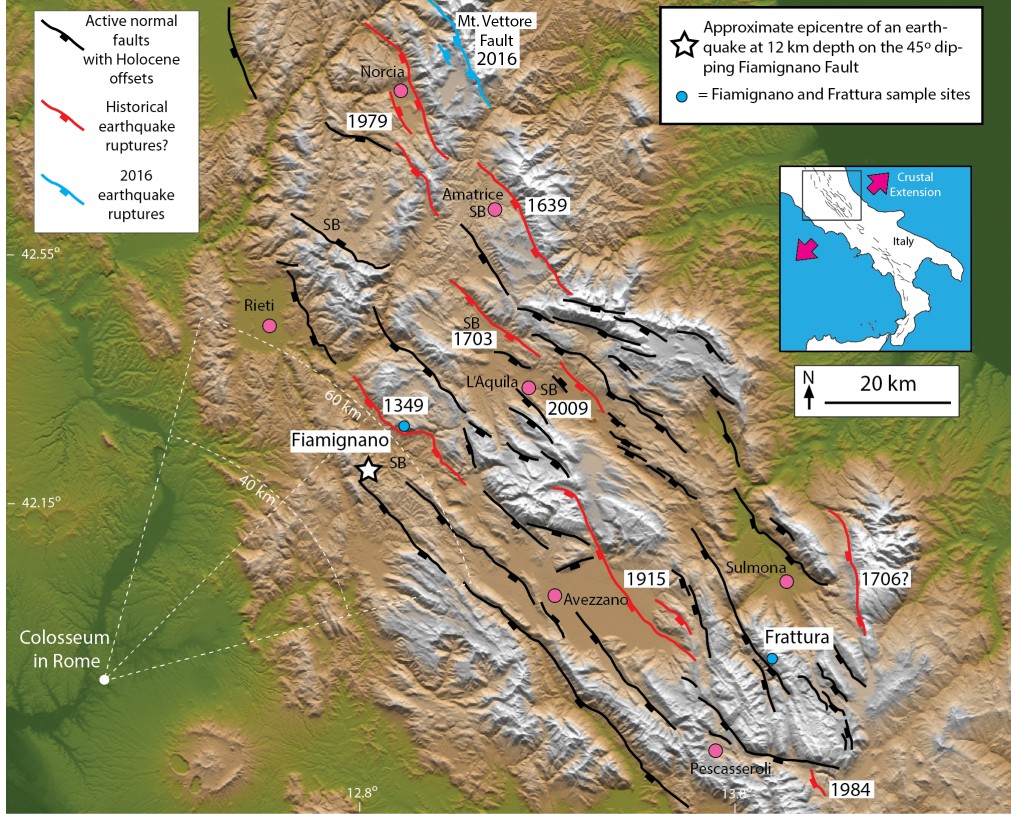

**Figure 8.** Location map of the central Apennines and the Fiamignano and Fraturra sample sites. Holocene active faults and historical ruptures adapted from (Roberts and Michetti, 2004; Pace et al., 2006; Cowie et al., 2017; Mildon et al., 2017)

intensity VIII damage is expected at distances of less than ~40–60 km of an epicenter from a $M_w$ ~ 6.0-6.5 event, and the Fiamignano fault is one of the few Apennines normal faults with that epicentral distance to Rome (see Fig. 8) thus making it a plausible source of the historic earthquake damage.

    Rome was damaged in earthquakes of intensity VIII during at least three earthquakes in the 5th, 9th and 14th centuries,
5  respectively. Other earthquakes damaged the city in 801, 1091, 1231, 1279, 1298, 1328 AD (Guidoboni et al., 2007). The Colosseum was damaged in 484 AD or 508 AD, based on a stone inscription where a person know as "Decius Marius Venantius Basilus," a prefect of the city, declares that he directly paid for restoration works after an earthquake; the uncertainty in age is because two Decius were consul, one in 484 AD and another in 508 AD. Damage consisted of collapse of the colonnades in the summa cavea (upper seating section for plebian spectators) with major damage to the arena and podium. Collapse of the outer
10  rings of the Colosseum is sometimes attributed to an 847 AD earthquake, when the nearby church of Santa Maria Antiqua was abandoned after earthquake damage (Vittori, 2015). An earthquake also damaged Rome on September 9, 1349 AD, one of 3 large earthquakes in the Apennines on that day. This earthquake is thought to be linked to the Fiamignano fault based on the observation that taxes were reduced in the vicinity of the Fiamignano fault due to misery and depopulation after the earthquake




(Guerrieri et al., 2002). Damage and collapse reported in 14th century accounts are summarized in (Galli and Molin, 2014), where the earthquake on that day is described as "the strongest seismic shaking ever felt in Rome" (Galli and Molin, 2014). It is postulated that there was an abrupt collapse of the southern external ring of the Colosseum during this earthquake, as observed in Fig. 9, because a bull fight was hosted there in 1332 AD, suggesting the Colosseum was intact, and an announcement that

5  the collapsed stones were for sale was made in the second half of the 14th century. Furthermore, an intact Colosseum can be seen on a coin from 1328 AD, whereas a damaged Colosseum is displayed on a 15th century image (Galli and Molin, 2014). Minor damage to the Colosseum also occurred in a 1703 AD earthquake (Vittori, 2015).

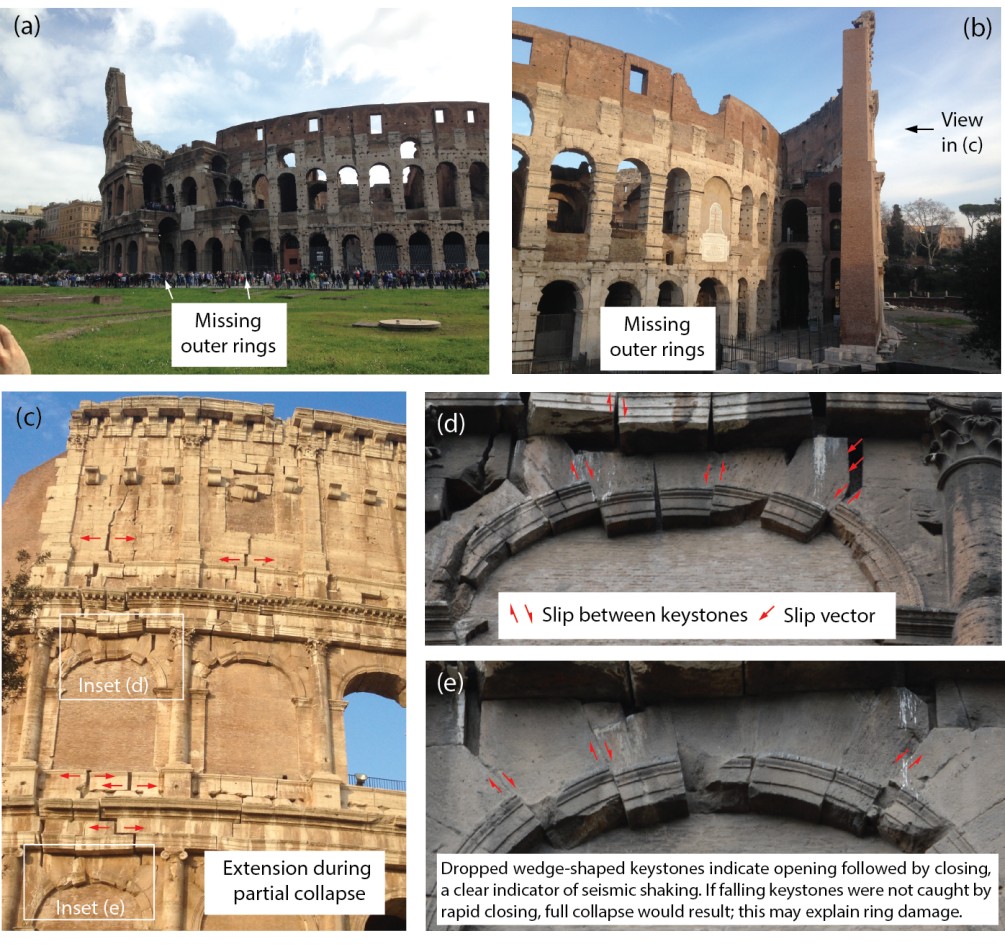

**Figure 9.** Damage to the Colosseum in Rome

One of the goals of this work is to investigate whether the Fiamignano fault is a candidate for the historical damage accounts in Rome based solely on $^{36}$Cl analysis. $^{36}$Cl data from the Fiamignano fault have previously been collected and analyzed in

10  (Cowie et al., 2017), where geomorphic and structural field mapping as well as laser and radar datasets were presented as evidence that their site was exhumed by tectonic slip as opposed to exhumation by erosion. Cowie et al. (2017) tried to infer an





earthquake history for the fault, and the results suggest rapid displacement between approximately 2000 years ago and 1349 AD. However, the results were biased by the modeling constraint that the most recent earthquake was enforced to be at 1349 AD and by the use of constant inter-event times between slip-rate change points and constant displacement sizes.

In our analysis, we use the same site parameter values and priors as in Sect. 4 except that we now use $d_{\max} = 300$, motivated by the large displacements for similar faults presented in (Wells and Coppersmith, 1994). We performed $1\,311\,016$ MCMC iterations of each chain and used the same number of chains, the same burn-in and thinning as in the synthetic case. A maximal PSRF value of 1.03 using two independent MCMC chains indicates convergence. The minimal weighted root mean squared error (wRMS) among all scenarios was 1.42. The time required for the offline calculations was ~15 minutes and the total time was ~12 days. In this work, we performed all numerical experiments on Intel Xeon E5-2680 v2 at 2.80 GHz (20 cores) with MATLAB 2016a.

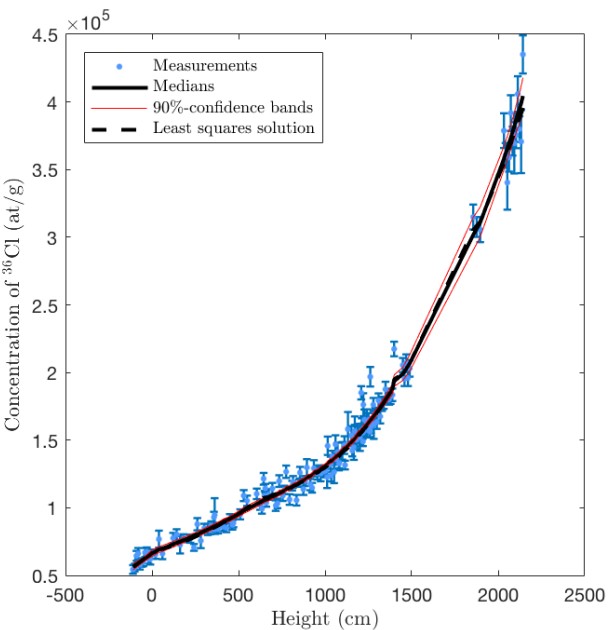

**Figure 10.** Fiamignano fault: measured and sampled $^{36}$Cl concentrations

The sampled posterior $^{36}$Cl concentrations fit well to the measurements as observed in Fig. 10. Our results agree with (Cowie et al., 2017) in that we find clear evidence of slip-rate variability and the slip-rate increasing through time before cessation of slip in the last 600-700 years. Although the present 28m offset in the plane of the fault occurred at an average slip rate of ~0.2 cm/yr, we find rapid slip of $1.0$–$1.4$ cm/yr centered around 1 ka, see Fig. 11. A detailed view for the period from 5ka to present is shown in Fig.12, where we observe that both of the two major earthquake events to have been observed in 847 AD and 1349 AD fall within this region of high intensity, see right top of Fig. 11, and potentially large displacements indicated by dark gray regions in the right bottom of Fig. 11. In fact, even the 801, 1091, 1231, 1279, 1298, and 1328 AD earthquakes fall within the



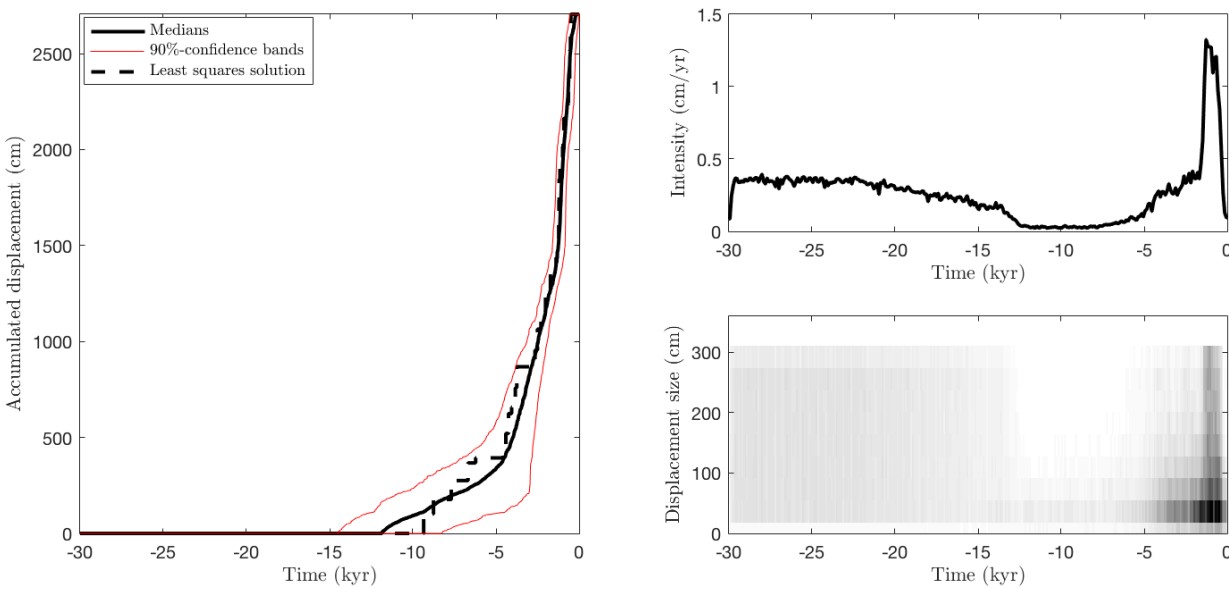

**Figure 11.** Fiamignano fault: the accumulated displacement (left), the mean earthquake intensity (right top) and an event scatter plot (right bottom) showing events from posterior samples (with grey scale to indicate frequencies).

high intensity region at times when our findings suggest relatively low displacement sizes. This deserves further investigation, and we hope it leads to palaeoseismic studies of offset late Holocene sediments on the Fiamignano fault that may be able to verify or refute these possibilities.

In conclusion, our findings suggest that slip is highly-clustered in time and that the Fiamignano fault is a plausible source of
the 847 AD and 1349 AD earthquakes associated with damage to the Colosseum and other ancient buildings known from the historical record.

## 5.2  Frattura fault

[36]Cl data for the Frattura fault have been collected and analyzed in (Cowie et al., 2017), where geomorphic and structural field mapping as well as laser and radar datasets were presented as evidence that their site was exhumed by tectonic slip as
opposed to exhumation by erosion. Unlike for the Fiamignano fault, the [36]Cl data were collected sparsely (15 samples), similar to the situation in our synthetic case with reduced amount of data shown. The site-specific parameters for the Frattura fault are $\alpha = 25$, $\beta = 53$, and $\gamma = 28$, fault scarp height $H_{sc} = 1570$, and trench depth $H_{tr} = 130$, the rock mean density $\rho_{rock} = 2.7$. Here the maximum displacement size is again chosen conservatively as $d_{max} = 300$; though we mainly expect displacements sizes less than $100\,\mathrm{cm}$ based on the data of (Wells and Coppersmith, 1994). As in the previous cases, we let $\rho_{coll} \sim \mathcal{U}([1.2, 1.8])$.
There are no known historical earthquakes for this fault consistent with records from towns and cities nearby, such as Sulmona and Pescasseroli, for which earthquake records extend back to Roman times. Indeed, the record is thought to be





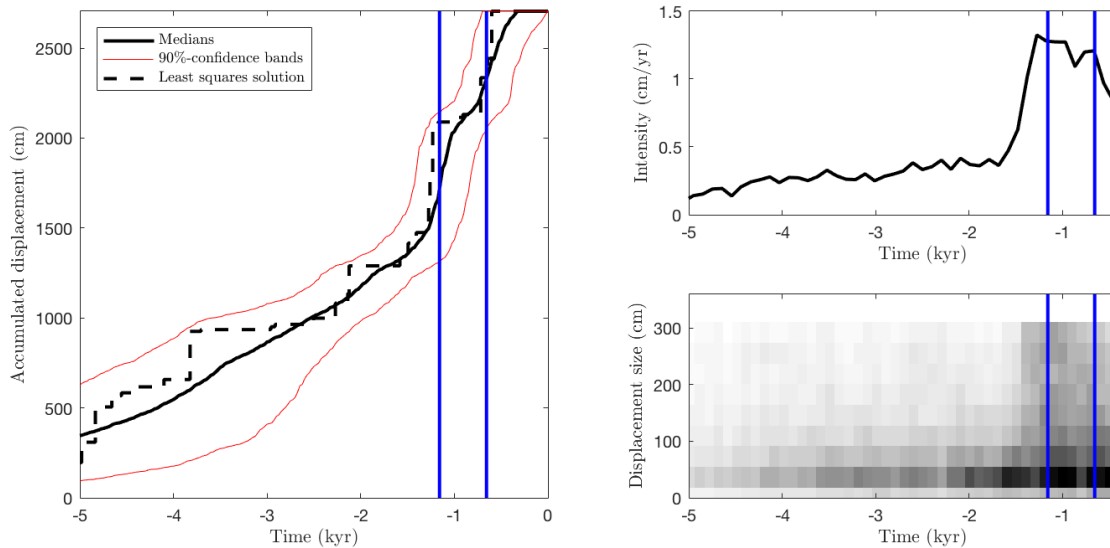

**Figure 12.** Fiamignano fault: detailed view of posterior displacement history and earthquake intensity. The vertical blue lines show the timings of two events that damaged the Colosseum.

complete since 1349 AD for magnitudes larger than $M_w$ 5.8; see (Guerrieri et al., 2002; Guidoboni et al., 2007; Pace et al., 2006; Roberts and Michetti, 2004) for discussions of the completeness period.

We performed $522\,560$ MCMC iterations of each chain and used the same number of chains, burn-in and thinning as in the synthetic case. A maximal PSRF value of 1.003 indicates convergence of the MCMC algorithm. The minimal wRMS among all scenarios was 1.78. The same settings are used for the offline calculations (~15 minutes) and the total time was ~3 days.

Figure 14 suggests a scarp age at approximately 19 ka consistent with expected slope stabilization ages, which may well be the marker for the time of the demise of the LGM shown in Fig. 1. After ~19 ka, Figures 14 and 15 indicate highly variable slip rates throughout the investigated time domain: initially a constant moderately-high slip rate until ~15 ka, a decreasing slip rate until ~8 ka, a sudden peak in slip from ~5–3 ka, and a low occurrence probability of earthquake events in the past ~2500 years. The lack of earthquakes in the past ~2500 years is consistent with historical earthquake records which show no major earthquakes in the vicinity.

The displacement histories of the Fiamignano and Frattura faults provide insights into slip-rate variability. The relatively short-lived bursts of high activity observed in our results occur at different times on faults in the same tectonic setting. This suggests that this is not a regional pulse of synchronous high slip, but probably related to the dynamics of interaction between each fault and its neighbors (Cowie et al., 2012), and that slip is highly-clustered in time, which has important implications for forecasting seismic hazard as discussed below.



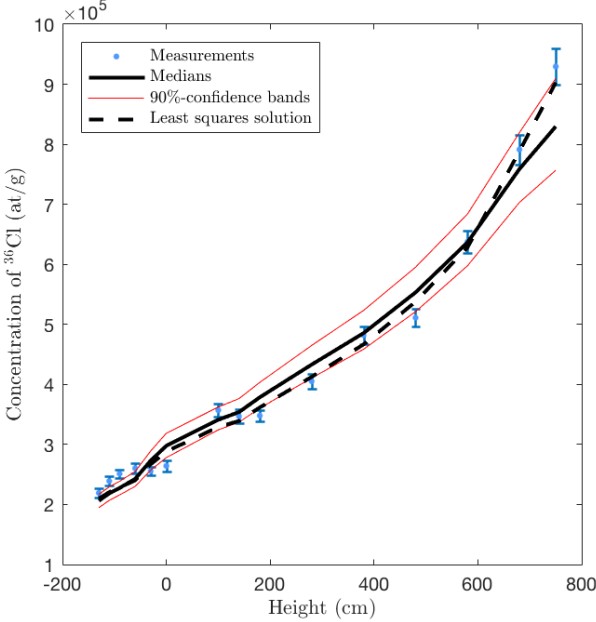

**Figure 13.** Frattura fault: measured and sampled $^{36}$Cl concentrations

## 5.3 Application to regional probabilistic seismic hazard assessment

Our results show that calculating probabilistic seismic hazard is considerably more challenging than previously thought. Current probabilistic seismic hazard calculations are based on the Brownian passage-time (BPT) distribution and require specification of the most recent earthquake time, the mean inter-event time, and the CV value (standard deviation of inter-event times divided by the mean inter-event time) to factor in variability in the inter-event time (Pace et al., 2016). However, our results show that in addition to the variability in inter-event times around a constant slip-rate, faults show heightened activity and quiescence over time periods lasting a few millenia relative to the longer term deformation rate. The differences in slip-rate between time of heightened activity (>1cm/yr) and quiescence (<0.1 cm/yr) are dramatic. These two timescales of slip-rate variability are not considered by current methods for calculating probabilistic seismic hazard (Pace et al., 2006, 2016; Tesson et al., 2016). To address this omission, we extend our more complex earthquake recurrence process – a Brownian motion with *time-varying* drift and noise – into the future, which enables us to sample more realistic next earthquake event times. More specifically, for each of our posterior samples, we continue the simulation of the corresponding Brownian motion sample path (such as that shown in Fig. 2) until another earthquake is generated. For this purpose, we use the sample-dependent values of the parameters describing the behavior of the Brownian motion: the average time $1/\Lambda$ between large-scale changes in slip activity, the small-scale recurrence variability $\tau$ and the hyperparameters $\alpha$ and $m$ that describe the distribution of average inter-arrival times in new slip activity periods. Thus, our approach is as a natural extension of the state-of-the-art methodology





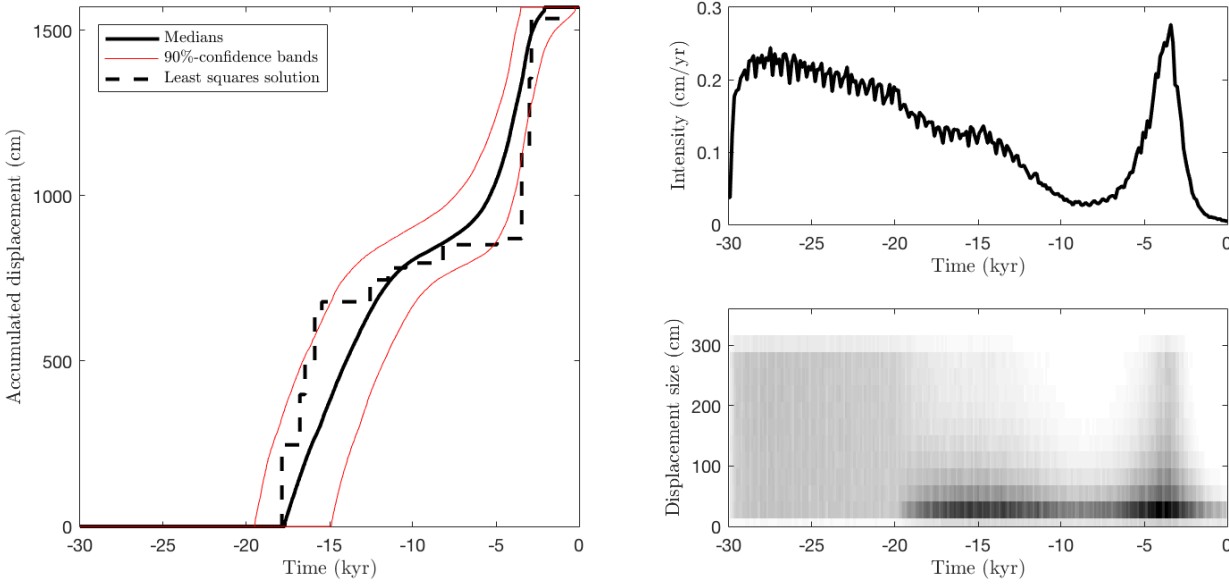

**Figure 14.** Frattura fault: the accumulated displacement (left), the mean earthquake intensity (right top) and an event scatter plot (right bottom) showing events from posterior samples (with grey scale to indicate frequencies).

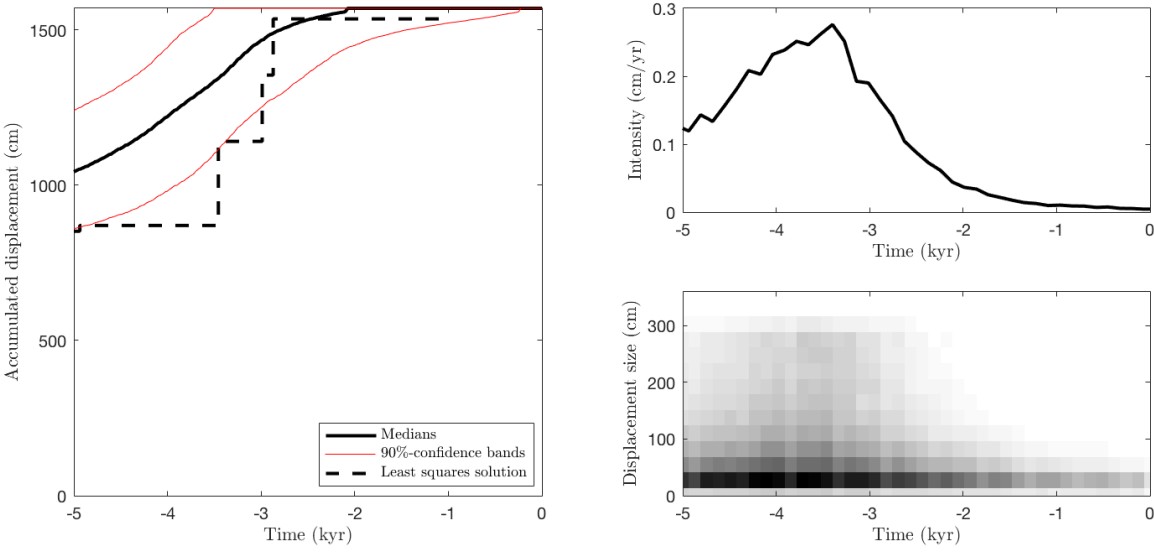

**Figure 15.** Frattura fault: detailed view of posterior displacement history and earthquake intensity.




that accounts for large-scale slip-rate variability and informs the resulting more complex model by the full displacement history at a given fault site.

We provide the posteriors of the next earthquake time for the Fiamignano and Frattura faults in Fig. 16. An important finding is that the values differ between the two faults, suggesting that with more data it may be possible to map such posteriors across entire regions such as that shown in Fig. 8. We expect site specific values for these posteriors to change on a length-scale of 20–30 km, the length of individual faults. Thus, our approach facilitates high spatial resolution seismic hazard mapping by implicitly including individual seismic sources (active/capable faults), the long-term slip-rates, and importantly the two timescales of slip-rate variability (millennial-scale heightened activity or quiescence, and inter-earthquake time variability).

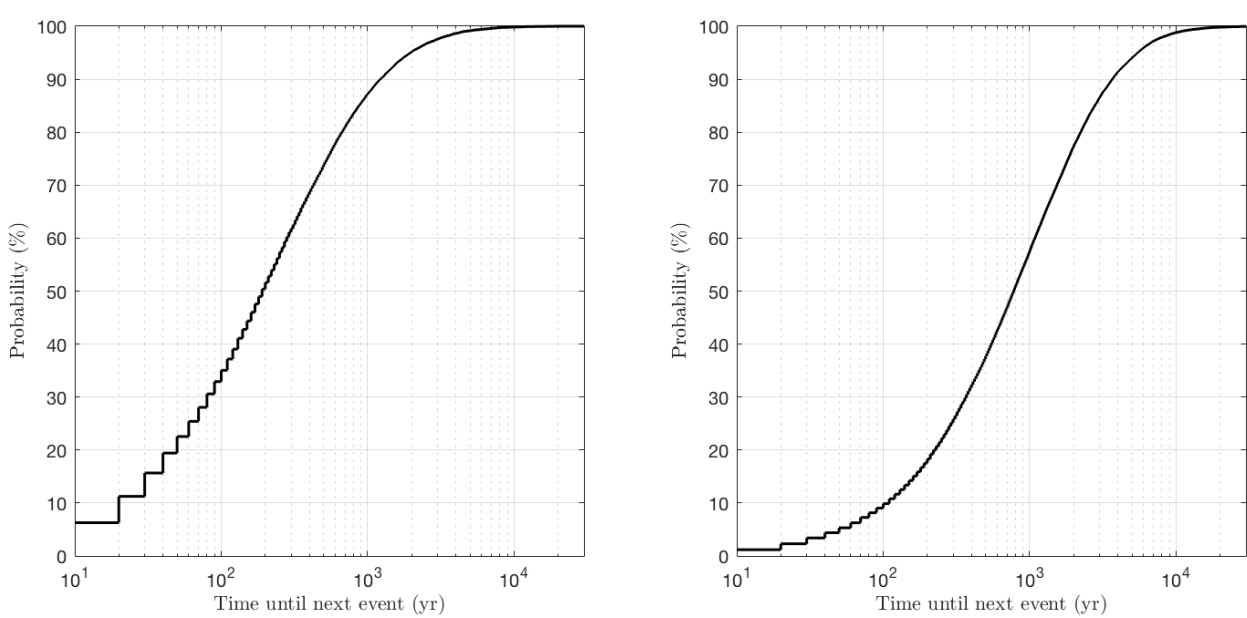

**Figure 16.** Posteriors of next earthquake time for Fiamignano (left) and Frattura (right)

However, a note of caution is that these results are probably only meaningful for near-future prediction. A physical basis that explains the cause of the slip-rate variability resulting from fault interaction would further improve probabilistic hazard analysis, as mentioned in (Tesson et al., 2016). During and after rupture, stress is resolved onto neighboring faults, and this is thought to produce temporal variation in slip-rate on faults that manifests itself in terms of temporal earthquake clustering (Scholz, 2010; Cowie et al., 2012; Mildon et al., 2017). Interaction allows the fault systems to share out the work associated with deforming a region between different faults so that only a few of the total number of active faults take part in regional deformation on millennial timescales (Cowie et al., 2012). Further work, including [36]Cl results from many faults in the same region, is needed to elucidate such interaction.



## 6   Conclusions

This work provides a validation of the $^{36}$Cl approach to inference of earthquake occurrence from the demise of the LGM until present. We propose a Bayesian MCMC method for the study of earthquake displacement histories, with applications to regional probabilistic hazard assessment. The method improves on the $^{36}$Cl modeling in (Schlagenhauf et al., 2010), the

Bayesian inference for $^{36}$Cl earthquake recovery in (Cowie et al., 2017), and the regional probabilistic hazard assessment in (Pace et al., 2006; Tesson et al., 2016). After demonstrating identifiability in the inverse problem through a synthetic case study, we present probabilistic earthquake displacement histories for the Fiamignano and Frattura faults in the Italian Apennines. We obtain highly variable slip-rates at both faults, in agreement with earlier studies of the region. At the Fiamignano fault, our findings suggest slip in earthquakes at times when the Colosseum and other ancient buildings in Rome were damaged.

Conversely, at the Frattura fault, our result is consistent with the fact that no large earthquakes were reported since Roman times.

*Code availability.*   The MATLAB code *Bayesian Earthquake Dating (BED) v1.0* of our Bayesian MCMC method for $^{36}$Cl earthquake dating and regional probabilistic seismic hazard assessment and the data required to reproduce our results are available as supplementary material and at GitHub https://github.com/beckjh/bed36Cl with version 1 also stored in the Zenodo repository https://doi.org/10.5281/zenodo.

1215594).

*Competing interests.*   The authors declare that they have no conflict of interest.

*Acknowledgements.*   This work was supported by NERC Directed Grant NE/J017434/1 "Probability, Uncertainty and Risk in the Environment", NERC Standard Grant NE/I024127/1 "Earthquake hazard from 36-Cl exposure dating of elapsed time and Coulomb stress transfer", NERC Standard Grant NE/E016545/1, "Testing Theoretical Models for Earthquake Clustering using 36Cl Cosmogenic Exposure Dating of

Active Normal Faults in Central Italy", and KAUST CRG4 Award 2584. We thank the many participants in these grants for discussions on earthquakes and $^{36}$Cl, although the views expressed in this paper are our own and any misconceptions are our sole responsibility.



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
