# Peer review of "Bayesian earthquake dating and seismic hazard assessment using chlorine-36 measurements (BED v1)"

_Geoscientific Model Development, 2018_

## Short Comment (SC1) · 18 Jul 2018

Earthquake dating poses the highest limitations to the seismic hazard assessment. The introduced Bayesian code BED v1 of this manuscript allows for the first time to account for several highly relevant uncertainties during the modelling of cosmogenic 36Cl data, resulting in more relevant modelling results. This runs in a very good combination with recent advances in the 36Cl sample preparation and measurement techniques, as well as new determined production rates.

In comparison to existing codes, BED v1 has several advantages:

- Significantly faster calculation

- The amount of coseismic offset per EQ can be treated as unknown instead of using rather arbitrary values

- The so far used arbitrary "pre-exposure component" is not required

- The consideration of uncertainties of the different production rates, soil density is easily accountable

The manuscript is very well written, easy to understand and the high-quality figures underline to content very well. I tested the code BEDv1 with the Matlab version 2016b, which runs without any problems. The handing is very easy out resulting figures easy to interpret and so demonstrative that they can be used directly. The extension, which directly applies a Bayesian regional probabilistic seismic hazard assessment, allows quick implications of the dataset results.

Here, I summarize some minor points to further improve the manuscript:

1. It would be helpful to provide more detailed supplementary information to better understand the code. For instance, some of the folders and parameters are not yet fully explained yet. E.g. the meaning of parameters.lambda_Sp and parameters.lambda_mu is probably not clear to most of the applicants. Furthermore, it would be helpful to differentiate "gamma" further. It is described as "upper eroded scarp dip (degrees)". Is this meant as being the footwall or the degraded scarp?

2. If feasible, the inclusion of some further parameters could allow a broader use of the code. As of now, the surrounding topographic shielding and the erosion of the fault plane is not accounted for (page 5 line 13: erosion does never stop completely). Maybe also uncertainties of the site geometry might be an interesting point for the future.

3. Fig. 2: description of y-axis is missing.

4. Fig. 3ff: It appears to be more intuitive to have the height on the y-axis.

5. Fig. 4b,7b: Intensity figure: please show the modelled and the true intensity of the input data for comparison. Like the blue dots used in Fig. 4c,7c.

6. The reference of the most recent 36Cl production rates is missing: Marrero et al. 2016, CRONUS-Earth cosmogenic 36Cl calibration, Quaternary Geochronology (e.g. page 3 line 21).

I feel that this code will be very useful particularly for the community of active tectonic research and I would like to emphasize that the manuscript represents a valuable contribution to GDM.

---

## Referee Comment (RC1) · J. T. Rougier (Referee) · 26 Jul 2018

**Reviewer's report:**

**Bayesian Earthquake Dating, GMD-2018-94**

I enjoyed this paper, but I will focus on section 3, as this is where I have some doubts about what the authors have done, and whether it is correct. These doubts stem from their explanation of MCMC at the bottom of p6, which is technically wrong. In line 25, it is the stationary distribution of the Markov chain which converges to the target, and there is a similar misunderstanding in line 29. This convergence implies that Cesàro means converge in mean-square to expectations; i.e. a sample from the chain can be used to estimate expectations. There is also some confusion in sec 3.2 which I will come back to below.

The authors have a statistical model with some parameters $\theta$, a stochastic process $\Pi$, and some parameters controlling the stochastic process, $\varphi$. They do not name $\Pi$ and $\varphi$ explicitly, but $\Pi$ is the stochastic process generating $\{(t_1, d_1), \ldots, (t_N, d_N)\}$. The authors will want to use a complicated $\Pi$, and this means, typically, that it will be simple to sample from $\Pi \mid \varphi$ but very hard to evaluate $p(\Pi \mid \varphi)$. $\Pi \mid \psi$ is complicated for two reasons. First, realistic earthquake models are much more complicated than marked Poisson processes; second, they will want to impose the constraints that $d_{\min} \leq d_i \leq d_{\max}$, and $\sum_{i=1}^{N} d_i = H_{sc}$. This would never happen by chance in an unconstrained simulation, and so it must be built-in to the prior, as explained at the bottom of p7.

I am skeptical of whether the authors are able to evaluate $p(\Pi \mid \psi)$ for their complicated model, which is a highly non-linear function of a Brownian motion (it is not clear in the MS whether or this is normal or geometric Brownian motion). I am also confused about the proposal described at the start of sec 3.2. I would like to point out that in a Metropolis-Hastings MCMC scheme they do not need to evaluate $p(\Pi \mid \psi)$, if they always propose $\Pi \mid \psi$ from the prior. The MH acceptance ratio is

$$\alpha = \frac{L(\theta', \Pi', \psi') \, p(\theta') \, p(\Pi' \mid \psi') \, p(\psi')}{L(\theta, \Pi, \psi) \, p(\theta) \, p(\Pi \mid \psi) \, p(\psi)} \times \frac{\operatorname{prop}(\theta, \psi) \, \operatorname{prop}(\Pi \mid \psi)}{\operatorname{prop}(\theta', \psi') \, \operatorname{prop}(\Pi' \mid \psi')}$$

where $L$ is the likelihood function, primes indicate the proposed values, and 'prop' is the proposal distribution, which may depend on $(\theta, \Pi, \psi)$. So if $\operatorname{prop}(\Pi \mid \psi) = p(\Pi \mid \psi)$, then the MH acceptance ratio simplifies to

$$\alpha = \frac{L(\theta', \Pi', \psi') \, p(\theta') \, p(\psi')}{L(\theta, \Pi, \psi) \, p(\theta) \, p(\psi)} \times \frac{\operatorname{prop}(\theta, \psi)}{\operatorname{prop}(\theta', \psi')}.$$

Now it is quite true that this chain will be slow to mix, if the likelihood is highly concentrated. But that is exactly why tempering is a good idea. Tempering is

a good trick for whenever we are forced to propose from the prior, owing to the difficulty of computing the probability density. I think there is an interesting message in this paper, which is that this method is applicable even though $\Pi \mid \psi$ is very complicated.

In their MH-MCMC scheme, $\text{prop}(\theta, \varphi)$ is a random walk, although I caution the authors to make sure that they include the Jacobian term if using a transformation: e.g., random walking in logs for non-negative parameters, or logits for bounded parameters, or else be clear about reflection at the boundaries. They might also consult, e.g., Andrieu and Thoms (2008), to implement an adaptive phase at the start of their chain, and to make cautious proposals using the principle components of the estimated posterior variance matrix. All of this needs to be stated in the MS.

The description of the proposal in sec 3.2 is confused. I believe that some issues have been resolved above, by always sampling $\Pi \mid \psi$ from the prior, and tempering. If the authors do not do this, then they will have to give explicit forms for $\text{p}(\Pi \mid \psi)$ and for $\text{prop}(\Pi \mid \psi)$, so that readers can implement the algorithm themselves. I do not understand the Reversible Jump part at all. $\Pi \mid \psi$ is a point process; the fact that the number of components varies is irrelevant.

In terms of diagnostics, the authors will need to demonstrate that their very intricate MCMC does indeed have the correct target distribution, using, e.g., the method of Cook et al. (2006); see also Dan Simpson's update at `http://andrewgelman.com/2018/04/18/better-check-yo-self-wreck-yo-self/`. This will also check that the authors are using a long-enough burn-in, and so separate Gelman-Rubin diagnostics are not required in the MS, although they may be helpful when setting-up the chain. As an aside, 2 independent chains is not nearly enough for Gelman-Rubin: 8 is much better.

Jonathan Rougier
University of Bristol
July 2018

**References**

Andrieu, C. and Thoms, J. (2008). A tutorial on adaptive MCMC. *Statistics and Computing*, 18:343–373.

Cook, S., Gelman, A., and Rubin, D. (2006). Validation of software for Bayesian models using posterior quantiles. *Journal of Computational and Graphical Statistics*, 15(3):675–692.

---

## Referee Comment (RC2) · B. Pace (Referee) · 27 Jul 2018

Review of the manuscript:

"Bayesian earthquake dating and seismic hazard assessment using chlorine-36 measurements (BED v1)" by Beck J. et alii

*The Authors present a MATLAB code, called BED v1, that using a Bayesan Markow-chain Monte Carlo approach account the relevant uncertainties involved in dating seismic events occurrences on fault planes by measuring the 36Cl abundance.*

*The subject of the article is of broad interest to the scientific community involved in this topic and can represent an original and significant contribution to fault-based seismic hazard studies.*

*The manuscript is properly organized and written clearly, the objectives are clear and the interpretations of the results are supported by the well presented data. Moreover, the code BED v1 runs without problem with the Matlab vers. 2016b and it is quite user-friendly.*

*Considering my skills I mainly focused my review on the application to regional probabilistic seismic hazard assessment and I suggest some minor revisions on this part to improve the paper and the interest of the seismic hazard modelers community. First of all the authors have to better explain, at the beginning of the paragraph, that only fault-based and time-dependent seismic hazard models, and not all the current probabilistic seismic hazard calculations, are mainly based on Brownian passage-time (BPT) distribution. Moreover, the choice of a BPT distribution try to take into account physically motivated models, where the probability of occurrence of the next earthquake on a single source cannot grow indefinitely but considers the possibility that, after an elapsed time close to the mean recurrence time of the characteristic earthquake, the probability follows a Poisson-like behaviour. The reason for this behavior can be linked to the fault system interaction effect. Furthermore, the coefficient of variation (CV) parameter takes into account the effects of the tectonic loading stress, the fault system geometry and the slip-rate variability (see for some details Visini and Pace, 2014, SRL). For these reasons, the assertion that such fault-based seismic hazard models do not consider the slip rate variability is not totally correct. In any case, in my opinion, the authors' proposed approach is very interesting and challenging. What is missing, from my point of view, is a comparison of the results with the "classical" BPT distribution, both in terms of next earthquake probability and, if possible, of probabilistic expected ground shaking (using a simple model). The probabilities shown in Fig. 16 seems to me very "high" but without a comparison with other approaches (e.g. FiSH approach, Pace et al., 2016, SRL) and without an application in terms of probabilistic seismic hazard maps (or curves) is not easy to understand the impact of the proposed methodology. The knowledge in terms of earthquake occurrences on*

*individual faults and of slip rate variability is growing fast and this manuscript and the related code can give another important incentive, and so some sensitivity tests to show the impact of different approaches I think are essential.*

*In conclusion, I suggest to the authors to improve the paragraph 5.3, in order to make the paper even more challenging, but overall I consider the manuscript and the related code an important step towards a next generation of fault-based seismic hazard models that includes fault interactions and slip rate variability.*

*Chieti 27/07/2018*

*Best regards,*
*Bruno Pace*

---

## Author Comment (AC1) · 21 Aug 2018

**Author Response to Short Comment 1 (S. Mechernich)**

We thank the referee for the detailed and constructive feedback. Point-by-point replies to the comments are provided below.

1) It would be helpful to provide more detailed supplementary information to better understand the code. For instance, some of the folders and parameters are not yet fully explained yet. E.g. the meaning of parameters.lambda_Sp and parameters.lambda_mu is probably not clear to most of the applicants. Furthermore, it would be helpful to differentiate "gamma" further. It is described as "upper eroded scarp dip (degrees)." Is this meant as being the footwall or the degraded scarp?

> `Lambda_sp` and `Lambda_mu` are explained on page 5 of the manuscript. We have added the explanation to the code as suggested. The parameter "gamma" is the dip of the upper slope of the footwall. We have changed the corresponding code comment.

2) If feasible, the inclusion of some further parameters could allow a broader use of the code. As of now, the surrounding topographic shielding and the erosion of the fault plane is not accounted for (page 5 line 13: erosion does never stop completely). Maybe also uncertainties of the site geometry might be an interesting point for the future.

> We agree that the modeling of the site geometry or the erosion process could be refined (e.g., we say "We approximate the effects of the LGM on erosion in a binary manner [...]" in the manuscript) and plan to do so in future versions of BED. For anyone wishing to do so on their own, we note that extensions or even replacements of the 36Cl simulation model can be done without significant changes to the MCMC algorithm (just add the line `settings.modelscarp=<yoursimulator>` and add additional parameters in the case study file).

3) Fig. 2: description of y-axis is missing.

> The y-axis represents the dimensionless value of the stochastic process that is used as a statistical model of earthquake recurrence.

4) Fig. 3ff: It appears to be more intuitive to have the height on the y-axis.

> Since this a matter of preference, we suggest that users simply swap axes using `figure(3);` `view([90,-90])` when required. This could be added at the end of `BED_plots.m` to swap axes permanently.

5) Fig. 4b,7b: Intensity figure: please show the modelled and the true intensity of the input data for comparison. Like the blue dots used in Fig. 4c,7c.

The plots of displacement size versus time plot and of accumulated displacement versus time contain the same information. The latter is best suited for a comparison between posterior and truth.

More precisely, the problem with displaying the truth in the intensity plot is the following: The posterior slip intensity as shown in the manuscript is defined as the *derivative* of the posterior mean displacement history. This produces a nice and well-defined result because the latter is relatively smooth (if enough posterior samples are used in its computation). However, since the true displacement history is a step function, its derivative is a sum of delta peaks, which is not suited for visualization or comparison to the posterior. Mathematically speaking, it is better to compare the displacement history because two very similar displacement histories may exhibit very different slip intensities. Think, for example, of a linearly growing displacement history and a staircase history with very small steps. The intensity of the former is uniform, the intensity of the latter is a series of peaks. The slip intensity should thus best be seen as a mere aid in the representation of the posterior displacement history.

6) The reference of the most recent 36Cl production rates is missing: Marrero et al. 2016, CRONUS-Earth cosmogenic 36Cl calibration, Quaternary Geochronology (e.g. page 3 line 21).

We include the reference as suggested.

---

## Author Comment (AC2) · 21 Aug 2018

**Author Response to Referee Comment 1 (J.T. Rougier)**

We thank the referee for the detailed and constructive feedback. Point-by-point replies to the comments are provided below.

1) the explanation of MCMC [...] is technically wrong

> We are aware that, mathematically speaking, what underlies MCMC is the property of the generated samples $(\theta_k)_{k=1}^{\infty}$ that the empirical measures $P_K := 1/K \sum_{k=1}^{K} \delta_{\theta_k}$ converge in some sense to the desired measure $P_{\theta|y}$ as $K \to \infty$. However, our goal is to keep the description of MCMC simple and targeted to a general geoscientific audience.
>
> We agree that we could improve the wording and change
>
> "To gain information about $P_{\theta|y}$, we employ an MCMC method, which generates samples that can be used to approximate statistical properties of $P_{\theta|y}$."
>
> to
>
> "To gain information about the posterior distribution $P_{\theta|y}$, we employ an MCMC method, which generates samples that, roughly speaking, behave as if they were drawn from the posterior and can therefore be used to approximate statistical properties thereof."
>
> One aspect of "behaving as if they were drawn from the desired distribution" is that the distribution of $\theta_k$ approaches $P_{\theta|y}$ as $k \to \infty$. We know that there is more to it (more on that in issue 2 below). However, if we attempted a more mathematically precise description and talked about convergence of empirical averages, it may still not be clear to a statistically inexperienced reader how that relates, e.g., to the convergence of kernel density estimates.

2) In line 25, it is the stationary distribution of the Markov Chain which converges to the target

> We assume the referee meant "In line 25, it is the stationary distribution of the Markov Chain which EQUALS the target [namely $P_{\theta|y}$]". We agree with this, but we do not see how it contradicts what we wrote. To be clear, what we meant in line 25 by "the sample distribution converges to the desired distribution as $k \to \infty$" is "$\mathcal{L}(\theta_k) \to P_{\theta|y}$ in some metric". We now make this more clear by saying "the distribution of $\theta_k$ converges to the [...]". This convergence does hold for aperiodic and $\phi$-irreducible MCMC chains (which we admittedly did not cover in the manuscript). The convergence of Cesàro means, i.e. of the empirical measures defined above, also requires verification of $\phi$-irreducibility. Since our chain is obviously aperiodic (the transition kernel satisfies $Q(x, \{x\}) > 0 \; \forall x$), we do not think it would be an advantage to talk about the convergence of Cesàro means. On the contrary, we believe delving into Markov chain theory and discussing mean-squared convergence of Cesàro means might deter possible readers. Since we are

not reinventing MCMC theory, we leave it to inclined readers to consider the cited references.

Again, the important part is that readers understand that the samples that we generate behave such that they represent the distribution $P_{\theta|y}$. One advantage of focusing on the convergence of distributions $\mathcal{L}(\theta_k)$ to $P_{\theta|y}$ is that it helps us make a case for parallel tempering. Indeed, the main result of the cited reference is that this convergence can be accelerated by parallel tempering.

3) The authors will want to use a complicated $\Pi$ and this means, typically, that it will be simple to sample from $\Pi|\psi$ but very hard to evaluate $p(\Pi|\psi)$ [...] I am skeptical of whether the authors are able to evaluate $p(\Pi|\psi)$ [we replaced all occurence of $\phi$ by $\psi$]

One of our proposals indeed consists in resampling the entire process from the prior, as suggested by the referee.

Additionally, we sometimes redraw only subintervals of the Wiener process that drives our earthquake generating process (see Equation (4) on page 7). By definition, this Wiener process is independent of the remaining components of the prior, so this can be done by a standard Brownian bridge construction (see lines 11ff on page 8).

Furthermore, we sometimes use local proposals to redraw the drift and volatility values of our earthquake generating process as well as the times at which those values change. Since these values together with their switch points constitute a marked Poisson process, the required Metropolis-Hastings ratio calculations can be found in the cited reference on "Reversible jump MCMC" (see line 10 on page 8 and issue 7 below).

4) "realistic earthquake models are much more complicated than marked Poisson processes"

While its drift and volatility coefficients are a marked Poisson process, our earthquake generating process itself is *not* a marked Poisson process. In any case, we agree that the model is not perfectly realistic. Nonetheless, our model is an improvement over the state-of-the-art Brownian passage time model, which is based on constant drift and volatility.

5) "it is not clear in the manuscript whether this is a normal or geometric Brownian motion"

Mathematically speaking, our earthquake generating process is an Itô diffusion (when conditioned on the drift and volatility coefficients). As such, it is driven by a standard Brownian motion. This is clear from Equation (4). In general, we don't think "Brownian motion" could seriously be used without qualifier to refer to a "geometric Brownian motion".

6) In their MCMC scheme, $prop(\theta, \phi)$ is a random walk, although I caution the authors to make sure that they include the Jacobian term if using a transformation

> Most of our proposals do not require a Jacobian term. For example, for simple production parameters, we randomly use either a proposal from the prior or a local random walk type proposal. Whenever the latter proposal proposes values outside of the bounded interval, the proposal is simply rejected. No log-transformations etc. are used. (As a sidenote, the fact that sometimes proposals are outside of the bounded prior intervals is one way to check $Q(x, \{x\}) > 0$, which we claimed above)

7) I do not understand the Reversible Jump part at all. $\Pi|\psi$ is a point process; the fact that the number of components varies is irrelevant.

> We agree that the fact that the number of components of the marked point process varies is irrelevant from a theoretical point of view. However, from a practical point of view, the corresponding calculations are not completely trivial, in particular concerning the careful treatment of Jacobians in the computation of MH ratios when draws are not from the prior distribution. The purpose of the cited work on "Reversible jump MCMC" is simply to present these calculations in theory and practice.

8) There is also some confusion in sec 3.2 which I will come back to below[...] The description of the proposal in sec 3.2 is confused.

> We are not sure which part is confusing. The referee states he believes "some issues have been resolved above, by always sampling $\Pi|\psi$ from the prior" and we hope this is the case since we indeed mostly use simple proposals from the priors and local random walk type proposals (see section 3.2 of the submitted manuscript). The only exception of non-trivial proposals are those for the marked Poisson process of drift and volatility values, which is why we added the Reversible Jump MCMC reference. In general, we tried to keep the description in the manuscript easy to understand and read and attached the MATLAB code of our implementation as reference for implementation details.

9) In terms of diagnostics, the authors will need to demonstrate that their very intricate MCMC does indeed have the correct target distribution, using [...] Cook [...] Simpson

> We thank the referee for bringing these very interesting verification tools to our attention. We implemented the test by Talts et al. (2018), the results of which are shown in Figure 1. However, since it is a very expensive test, we could only generate 200 samples in the month until this revision deadline, and each was only run until a point where the GR diagonistic was around 1.5 (200 samples means 200 synthetic test cases like the one presented in the manuscript). We changed the sentence

[Figure]

Figure 1: MCMC implementation verification following Talts et al. (2018): We generated 200 synthetic truths from the prior distribution and ran our MCMC algorithm for each. We used 10% burn-in and thinned each chain by factor of 250 to obtain quasi-independence. This left us with 42 posterior scenarios for each of the 200 runs of the MCMC algorithm. The distribution of the rank of the truth among these 42 scenarios is plotted above (where the ranking is based on the accumulated displacement at $-7$ka, a time at which the average fault has reached approximately half of its final height). A $\chi^2$-test on uniform distribution of these ranks returned the $p$-value 0.54 (which is considerably too large to reject the null hypothesis of uniform distribution).

"We verify global convergence with the diagnostic of Gelman and Rubin (1992)."

in the introduction to

"We verify correct implementation of our MCMC algorithm by the test of Talts et al. (2018) and global convergence with the diagnostic of Gelman and Rubin (1992)."

We prefer to not include Figure 1 in the manuscript. In the end, this is just one of many possible implementation tests and we believe that the best way to verify software is to make it open and free. For example, even with the implementation of the MCMC algorithm verified, the simulation of 36Cl concentrations and the sparse grid surrogate for attenuation factors, neither of which are much easier to implement than the MCMC algorithm itself, can only be verified by applying our algorithm to real case studies or by looking at its source code (or by a handful of sanity checks, such as case-wise comparison with alternative simulators, monotonocity/positivity checks, application to trivial earthquake scenarios with obvious solution properties, etc., which we did perform but didn't include in the manuscript either). We do prefer to keep the much cheaper GR diagnostic for simple convergence testing. To use 8 instead of 2 independent chains, users may simply add `settings.group_size=8` to the case study text file.

---

## Author Comment (AC3) · 21 Aug 2018

**Author Response to Referee Comment 2 (B. Pace)**

We thank the referee for the detailed and constructive feedback. Point-by-point replies to the comments are provided below.

1) First of all the authors have to better explain, at the beginning of the paragraph, that only fault-based and time-dependent seismic hazard models, and not all the current probabilistic seismic hazard calculations, are mainly based on Brownian passage-time (BPT) distribution.

   We changed "Current probabilistic seismic hazard calculations are [...]" to "Fault-based and time-dependent seismic hazard models are [...]".

2) The assertion that such fault-based seismic hazard models do not consider the slip rate variability is not totally correct.

   We believe this is related to the statement "However, our results show that in addition to the variability in inter-event times around a constant slip-rate, faults show heightened activity and quiescence over time periods lasting a few millenia relative to the longer term deformation rate. The differences in slip-rate between time of heightened activity (>1cm/yr) and quiescence (<0.1 cm/yr) are dramatic. These two timescales of slip-rate variability are not considered by current methods for calculating probabilistic seismic hazard (Pace et al., 2006, 2016; Tesson et al., 2016)."

   This is not stating that such fault-based seismic hazard models do not consider the slip rate variability at all, but that such models are not explicitly accounting for slip rate variability on both timescales. To our knowledge, our approach is the first of this kind, if this is not true then we would be grateful if you can provide references.

3) What is missing, from my point of view, is a comparison of the results with the classical BPT distribution, both in terms of next earthquake probability and, if possible, of probabilistic expected ground shaking (using a simple model). The probabilities shown in Fig. 16 seems to me very high but without a comparison with other approaches (e.g. FiSH approach, Pace et al., 2016, SRL) [...] is not easy to understand the impact of the proposed methodology.

   We have now included a comparison in our examples with the standard BPT approach using the MCMC samples based on the earthquake records produced by our MCMC algorithm, because there is to our knowledge not any earthquake record for a single fault that is not too sparse to estimate the parameters of the BPT distribution and that will typically underestimate the probabilities. In Figure 1, we can observe the difference between the results.

To the best of our knowledge some damaging earthquakes produce slips on the fault as small as 10 cm. However, if one is convinced that damaging earthquakes have a minimal size of 50 cm then obviously the MCMC will result in scenarios with less earthquakes and thus in lower hazard probabilities for the next earthquake. To illustrate this fact, we have also included a run of the MCMC algorithm for the slip size 50 cm.

[Figure]

Figure 1: Posteriors of next earthquake time for Fiamignano (left) and Frattura (right) with BED run using $d_{min} = 10$, $d_{min} = 50$, and with the standard BPT distribution with $T_{mean}$ and $CV$ from the MCMC samples obtained from the BED run using $d_{min} = 10$.

4) ...without an application in terms of probabilistic seismic hazard maps (or curves) is not easy to understand the impact of the proposed methodology.

It is beyond the scope of the paper to produce hazard maps as there are many different ways to attenuate shaking with distance in the literature, and we do not want to restrict our attention to a particular one. Instead, the reader is provided with enough information to make their own seismic hazard map from the results produced by our method. The method is applied to individual faults and the probabilities for the next event time can be integrated into creating probabilistic seismic hazard maps in the same way as standard BPT based methods.